# Design of a Half-Mode Substrate-Integrated Waveguide (HMSIW) Multimode Resonator Bandpass Filter Using the Minkowski Fractal for C-Band Applications

**DOI:** 10.3390/mi15121440

**Published:** 2024-11-28

**Authors:** Nitin Muchhal, Abhay Kumar, Nidhi Tewari, Samriti Kalia, Shweta Srivastava

**Affiliations:** Department of ECE, Jaypee Institute of Informaton Technology, Noida 201309, UP, India; abhay.kumar@jiit.ac.in (A.K.); nidhi.tewari@jiit.ac.in (N.T.); samriti.kalia@jiit.ac.in (S.K.); shweta.srivastava@jiit.ac.in (S.S.)

**Keywords:** bandpass filter, SIW, fractal, DGS, C-band

## Abstract

A substrate-integrated waveguide (SIW) bandpass filter (BPF) with extraordinary selectivity and an adequate upper stopband for C-band Satellite Communication (SATCOM) applications is proposed in this paper. The design comprises comb-shaped slots engraved on a half-mode SIW (HMSIW) that constitute a multimode resonator (MMR). Its performance is further ameliorated by applying the first and second iterations of the Minkowski fractal curve in the ground plane as a defected ground structure (DGS). The Minkowski fractal has advantages in terms of better bandwidth and miniaturization. The filter is first simulated using the commercial full-wave electromagnetic simulator HFSS v19 and then fabricated on a 0.062′′ (1.6 mm) FR4 with dielectric constant ε_r_ = 4.4. The measured results are comparable with the simulated ones and demonstrate that the BPF has a resonant frequency (f_0_) of 4.75 GHz, a 3 dB bandwidth of 770 MHz (fractional bandwidth of 21.4%), an insertion loss of 1.05 dB, and an out-of-band rejection (in the stopband) of more than 28 dB up to 8 GHz, demonstrating a wide and deep stopband. Using the multimode resonator (MMR) technique, a wide bandwidth has been achieved, and by virtue of using half-mode SIW (HMSIW), the proposed BPF is compact in size. Also, the fractal DGS aids in better stopband performance.

## 1. Introduction

Bandpass filters (BPFs) have been extensively used in satellite communication applications and are indispensable in abundant microwave and radio frequency (RF) applications. While designing a BPF, preferred characteristics comprise compact size, minimal insertion loss, wide upper stopband, better filtering performance, and reduced cost [1]. With its high Q factor, greater power-handling capability, ease of incorporation with planar structures, and other advantages over standard microstrip and waveguide geometries, the substrate-integrated waveguide (SIW) [2] method is extensively used. Nonetheless, since its width is larger, one of the major concerns is diminishing the width of the SIW structure. A number of SIW miniaturization techniques were thoroughly investigated in [3].

In contemporary times, a significant quantity of research has been conducted to design a miniaturized selective SIW BPF with good stopband performance for various bands of satellite communication applications. A compact SIW bandpass filter in the C-band was proposed in [4] by etching D-shaped resonators acting as complementary split-ring resonators (CSRRs) with face-to-face alignment on the SIW’s top layer, achieving a resonant frequency of 6.11 GHz. CSRRs are planar metamaterial structures that can yield negative values of effective permittivity characteristics of a material. A BPF using an HMSIW design based on transverse slots was proposed by [5], in which the frequency was tuned by altering the slot length. Various broadside-coupled complementary split-ring resonator (BC-CSRR) pairs with diverse orientations for the design of SIW BPFs were investigated in [6]. Further, a modified novel BC-CSRR pair was proposed in order to increase the rejection bandwidth. A compact HMSIW C-bandpass filter having a center frequency of 6.5 GHz was proposed and designed by [7]. The design consisted of C-shaped slots acting as a multimode resonator and two dumbbell-shaped DGSs on the ground plane to enhance the filter performance. A fourth-order cross-coupled bandpass filter was proposed and realized by [8] with QMSIW cavities and an S-shaped slot-coupling structure having a center frequency of 5.47 GHz. A selective BPF for Ku/K bands was proposed and designed by [9] using folded SIW (FSIW) technology integrating an E-shaped slot in the central septum. In recent times, higher-order BPFs for satellite applications were proposed by [10] using multiple single-mode resonators and combline resonators. A compact and selective composite microstrip BPF embedded with a pair of quarter-wavelength open-circuited stubs and a spurline was proposed by [11]. Most recently, a dual-band SIW BPF multimode resonator with overlapping modes was proposed by [12]. To control the overlapping modes, two types of slotlines and a pair of via perturbations were introduced. An SIW BPF with an adaptable passband using assembled multimode resonant printed circuit boards (PCBs) was proposed by [13]. To achieve the requisite filtering response with a different number of passbands, three distinct PCBs with perturbed modes were used.

This work realizes a miniaturized selective three-pole bandpass filter by integrating a Minkowski fractal geometry (as a defected ground structure) with a comb-shaped slotted half-mode SIW. The idea of the half-mode SIW BPF was introduced by [14], which diminishes the size of the original SIW structure by practically half. Fractal geometry [15] has been extensively used in electromagnetics to improve performance and miniaturize systems by leveraging fractals’ space-filling characteristics in passive microwave circuits and antenna design. The designs are simulated using the EM simulator HFSS v.19 software. Hence, this design is completely novel as it combines features of a multimode resonator, half-mode SIW, and fractal DGS for C-band applications. Such a design is proposed and validated for the first time.

A brief outline of the proposed filter design process is as follows: For the proposed BPF design, first an HMSIW for the desired cutoff frequency is designed. Thereafter, a comb-shaped MMR structure is introduced on the upper metallic plane and analyzed. Further, to enhance the stopband performance, the first and second iterations of the Minkowski fractal DGS are introduced in the ground plane. After successful simulation, the BPF is fabricated and tested for validation. The structure of this paper is as follows: Section 1 elucidates the introduction and literature survey part of this paper. Section 2 outlines the design of the comb-shaped slotted half-mode SIW (HMSIW) filter. Section 3 explicates the design and analysis of the BPF based on the first iteration of the Minkowski fractal curve as the DGS. Section 4 discusses the design and analysis of the BPF based on the second iteration of the Minkowski fractal curve. Section 5 elucidates the fabrication and measured results of the proposed design. Section 6 presents the conclusion of the proposed work followed by references.

## 2. Design of a Comb-Shaped Half-Mode SIW (HMSIW) with Comb Slots

Figure 1 shows the basic HMSIW structure with feed designed on an FR4 substrate with Ɛ_r_ = 4.4, tan δ = 0.02, and a thickness of 1.6 mm. Further, comb slots are embedded on the upper plane of the HMSIW. The basic SIW acts as a high-pass filter. The advantage of the chosen design method is that it is simple to design and economical. It has the disadvantage that the SIW needs vias, and hence, more precision is needed. Using the standard SIW [16] and HMSIW [17] design equations for cutoff frequency f_c_ = 8 GHz, the various parameters are found to be as follows: HMSIW length L_HM_ = 12 mm, microstrip line length L_M_ = 3.5 mm, microstrip line width W_M_ = 0.65 mm, width of HMSIW W_HM_ = 5.2 mm, via diameter D = 0.75 mm, pitch P = 1.0 mm, and X = 1.4 mm. The dimensions of the comb slot for optimum performance are found as follows: thickness T = 0.25 mm, interspacing between consecutive combs B = 1.25 mm, and height A = 4.75 mm.

According to the Babinet principle, a slot is reciprocal to its metallic structure and can be represented by a corresponding model of inductance and capacitance [18]. The resonant circuit’s L and C create a passband, which causes this slotted structure to produce a rejection band at the desired frequency.

The spacing between two comb-shaped slots is found using parametric analysis as shown in Figure 2. It is evident that as the distance (S) between slots increases, the selectivity of the filter degrades, although there is a slight improvement in the loss, so the optimum result (for acceptable insertion loss and roll-off rate) is found for S = 1.2 mm.

Figure 3 elucidates the simulated S21 response of the multimode resonator filter for weak coupling, from which it is evident that this filter creates two resonant modes to form the passband. Moreover, from the mode chart shown in Figure 4, it can be realized that by varying the stub length (L), transmission zeros’ (TZs’) and resonant modes’ locations change. The TZ frequency swings down with an upsurge in the stub length. TZ2 is more affected by this than TZ1. This effect is more perceptible for TZ2 than TZ1. Also, two distinct resonant modes (5.8 and 6.65 GHz) form the passband for the optimal value of slot length (L) = 3.4 mm.

Figure 5 elucidates the frequency response of the BPF. The BPF possesses a resonance frequency of 6.45 GHz with an insertion loss of 0.45 dB and a maximum return loss of 18.5 dB. The bandwidth is evaluated to be 0.78 GHz (6.12 GHz–6.90 GHz).

## 3. Band Pass Filter Based on the First Iteration of the Minkowski Fractal Curve

Fractal geometries were first proposed by Mandelbort in 1953 [19], and since then, they have been broadly used in diverse engineering domains. A fractal is a geometric structure that fills up space and has the same statistical properties throughout all of its parts. Fractal shapes can deliver big surface areas in a restricted amount of space as they are created through a recursive process. A generator curve can be designed using fractal geometry using the Iteration Function System (IFS) [20]. It is an elementary unit for creating the fractal structure’s subsequent iterations. The IFS can be written using matrices given in (1).
(1)$$H\;\left[\begin{array}{c} X\\ Y \end{array}\right]=\left[\begin{array}{cc} a & b\\ c & d \end{array}\right]\left[\begin{array}{c} X\\ Y \end{array}\right]+\left[\begin{array}{c} e\\ f \end{array}\right]$$

Two elements, e and f, control the linear translation, while four elements, a, b, c, and d, control the scaling and direction. Presume that the sequences H1, H2, …, Hn are linear affine transformations. Through a series of transformations on H (A), it can be represented by means of Equation (2). H is called the Hutchinson operator and is used to obtain the fractal geometry.
(2)$$H\;(A)={⋃}_{n=1}^{N}\;H_{n}\;(A)$$

Figure 6 illustrates the first three iterations of the Minkowski fractal curve, and Figure 7 shows the first iteration of the Minkowski curve with dimensions as Li = Wi = 1.45 mm and Di1 = 1.25 mm. Figure 8 depicts a comb-slotted HMSIW BPF with the first iteration of the Minkowski curve on the ground plane, which acts as the defected ground structure (DGS). A DGS [21] is a defect or slot in the ground plane of a microwave planar circuit that can be used to enhance its performance.

Figure 9 elucidates the frequency response of the BPF with the first iteration level of the Minkowski curve. With a maximum return loss of 20.4 dB, the BPF resonates at 5.85 GHz. The insertion loss is 0.58 dB with a bandwidth of 1.08 GHz (5.32 GHz–6.40 GHz). It is evident from the S parameter response that the addition of a DGS caused the lower frequency to shift to 5.3 GHz. The slow wave effect [23] is the reason behind the increase in the resonator’s effective electrical length that lowers the cutoff frequency causing size diminishment by 21.4%. In other words, the defect etched into the ground plane of a microwave circuit disturbs its current distribution, affecting the transmission line’s capacitance and inductance. The augmented capacitance and inductance cause a slow-wave effect, enhancing the effective electrical length of the patch. This results in a decrease in the resonant frequency causing miniaturization [23]. Also, it is evident that there is a slight enhancement in the bandwidth as parasitic capacitance is produced by the existence of the ground plane defect as it upsurges the fringing field. This parasitic capacitance is the reason for the increased coupling amid the metallic top plane and the bottom ground plane, which produces a wider bandwidth [24].

## 4. Band Pass Filter Based on the Second Iteration of the Minkowski Fractal Curve as DGS

Figure 10 demonstrates the second iteration of the Minkowski curve with dimensions Li2 = Wi2 = 0.45 mm and Di2 = 0.37 mm. Figure 11 shows the structure of the proposed BPF with comb slots on the upper metallic layer and the second iteration fractal DGS on the ground plane.

The equivalent circuit of the proposed design with parameter values is shown in Figure 12. The SIW structure acts as a two-wire transmission line; therefore, inductor L_v_ symbolizes metalized vias. A parallel resonator tank circuit is analogous to defective ground structures (DGSs) etched on the ground plane, which acts as a band stop filter [25]. Hence, the fractal DGS resonator is represented by Lr and C_r_, where the values of parameters Lr and Cr can be found using [26]. The electric and magnetic coupling due to comb-shaped slots at the input and output are represented by L_c_ and C_c_.

The simulated scattering response of Minkowski’s fractal-based filter is shown in Figure 13. It is clear from the response that the extended electrical length has caused the lower frequency to shift to 4.1 GHz. Furthermore, it can be shown that the proposed filter’s 3 dB bandwidth of 1.05 GHz (4.1 GHz–5.15 GHz) has been increased by the inclusion of the DGS. Moreover, it attains a deep and wide-ranging stopband (>−28 dB) until approximately 2f_0_. The insertion loss is found to be 0.72 dB. The roll-off rate at the higher and lower edges of the passband is 68.7 dB/GHz and 73.4 dB/GHz, respectively. Also, it is evident that it produces three transmission zeros at 3.17 GHz, 6.2 GHz, and 6.9 GHz, respectively, which augments selectivity and ameliorates the upper stopband performance. Sub-structures of the resonators come nearer, and the included length increases as the iteration degree rises while maintaining a constant side length. It produces capacitive coupling, which enhances the skirt properties of the filter at both edges [27]. This also results in an improvement in the filter’s stopband performance [28].

The equivalent circuit was validated using the Advanced Design System (ADS ver. 2019) software tool, which was used to design an equivalent circuit with RLC (as shown in Figure 12) and optimized it to obtain a nearly similar S parameter response, also shown in Figure 14, which compares the response of EM HFSS simulation and circuit simulation.

Figure 15 depicts the S21 response of the comb-slotted MMR filter with a DGS under weak coupling. Figure 16 illustrates the mode chart of the projected MMR filter with a DGS. It is evident from the mode chart that the desired passband is formed by the resonant modes (at f = 4.1, 4.25, 4.8, and 5.2 GHz) that shift down to lower frequencies after DGS is incorporated.

Figure 17 portrays the current distribution in (a) the passband and (b) the stopband of the proposed BPF.

## 5. Fabrication and Results

To corroborate the outcome, the proposed filter was fabricated with FR4 substrate material, which had a depth of 1.6 mm, a relative dielectric constant of 4.4, and a tan δ = 0.02. FR4 substrate is readily available and economical. The filter was fabricated using Accurate’s CNC PCB prototyping machine, IN, USA and tested using a vector network analyzer in the JIIT Microwave Research Laboratory. Figure 18a,b portray the image of the upper and lowermost sheet of the BPF with complete dimensions of 19.5 mm (length) × 6.5 mm (width). A one INR Indian coin (INR) is used for size reference.

The Anritsu S 820E vector network analyzer (CA, USA) available in our microwave research laboratory was utilized to measure the scattering properties of the manufactured filter. A two-port SOLT (short open load and thru) calibration was carried out to take into consideration the losses brought on by the cable joining the VNA and the DUT. Figure 19 compares and illustrates the outcomes of the HFSS simulation with the measured data. With a center frequency of 4.7 GHz, it displays the observed passband of the filter, which ranges from 4.1 to 4.87 GHz with a 3 dB bandwidth of 0.77 GHz and a 3 dB fractional bandwidth (FBW) of 21.4%. In the passband, the insertion loss is 1.15 dB, while the maximum observed measured return loss is 18.8 dB. The measured result shows that it attains good attenuation in the upper stopband. The measured roll-off rates at the passband’s upper and lower edges are 62 dB/GHz and 68.5 dB/GHz, respectively. Also, three transmission zeros are produced at frequencies of 2.65 GHz, 6.12 GHz, and 6.87 GHz, which augments the selectivity of the response. As shown in Figure 18, the out-of-band rejection in the stopband is >28 dB till 8 GHz. Figure 19 shows that there is worthy conformity amid the simulated and measured result, except for a petite swing in measured return loss and roll-off rate, which may be affected by the tolerance in the fabrication. There is a slight deviation between simulated and measured results. This error may be attributed to fabrication error and/or during measurement due to bending of the cable, connector loss, etc. It may be due to imprecise permittivity in the simulation.

Figure 20 depicts a comparison of the simulated and measured group delay responses, which elucidates a nearly flat response across the desired frequency band.

A performance investigation including the Q factor [29] of some recently reported SIW bandpass filters for C-band applications is encapsulated and compared in Table 1.

From the above table, it can be established that in [4], an HMSIW filter is made using D-shaped resonators. It has reasonably good stopband performance but suffers from low selectivity. The BPF proposed in [5] is designed using the HMSIW methodology consisting of transverse slots. Although the design is simple, it suffers from a large size and poor selectivity. The filter proposed in [6] using the broadside coupled CSRR technique has a smaller size but has the disadvantage of very high insertion loss. The filter proposed in [7] is made using HMSIW with DGS technology. It has the disadvantage of a larger size and poor stopband performance. The filter proposed in [8] is designed using quarter mode SIW techniques with S-shaped slot tech, but it suffers from a very large size. Consequently, the proposed filter with a small size, low loss, and better frequency response can be useful for various C-band applications.

## 6. Conclusions

In this paper, a half-mode SIW bandpass filter is analyzed, simulated, fabricated, and tested. To obtain a wide bandwidth, the filter is loaded with comb-shaped slots, which also act as multimode resonators. To miniaturize the size and improve the selectivity as well as the stopband performance of the filter, it is integrated with the Minkowski fractal, which acts as the DGS. The proposed filter achieves diminutive size, low insertion loss, and optimum bandwidth (3 dB FBW 23.4%) with good stopband performance. Since the proposed filter has the merit of good filtering properties, it can be useful for various SATCOM within C-band applications.

## Figures and Tables

**Figure 1 micromachines-15-01440-f001:**
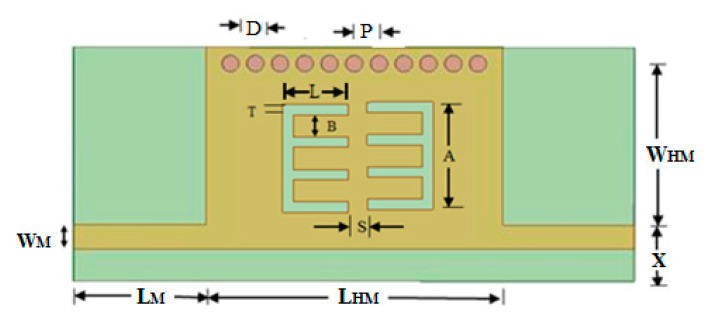
Half-mode substrate-integrated waveguide with a comb slot.

**Figure 2 micromachines-15-01440-f002:**
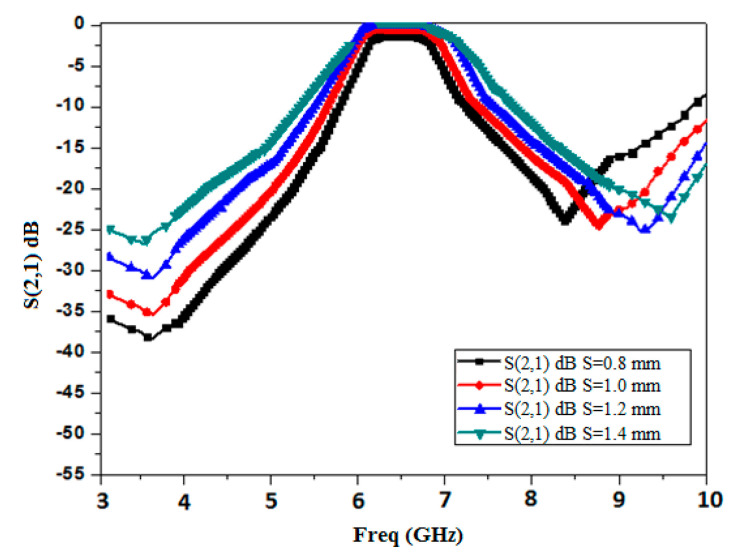
Parametric analysis of the transmission response of the comb-shaped curve with respect to edge spacing, S (in mm).

**Figure 3 micromachines-15-01440-f003:**
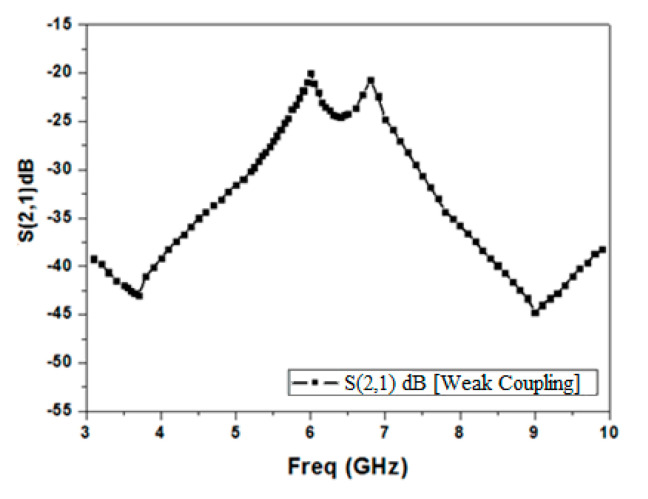
Simulated S21 response of a comb-slotted MMR filter under weak coupling.

**Figure 4 micromachines-15-01440-f004:**
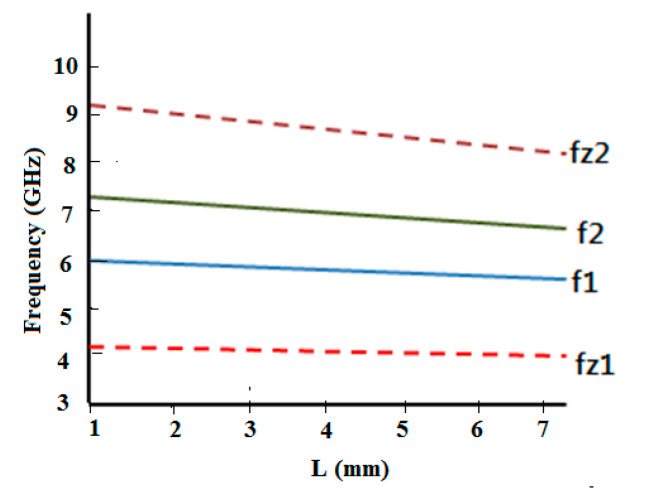
Mode chart of a comb-slotted MMR filter for different values of L.

**Figure 5 micromachines-15-01440-f005:**
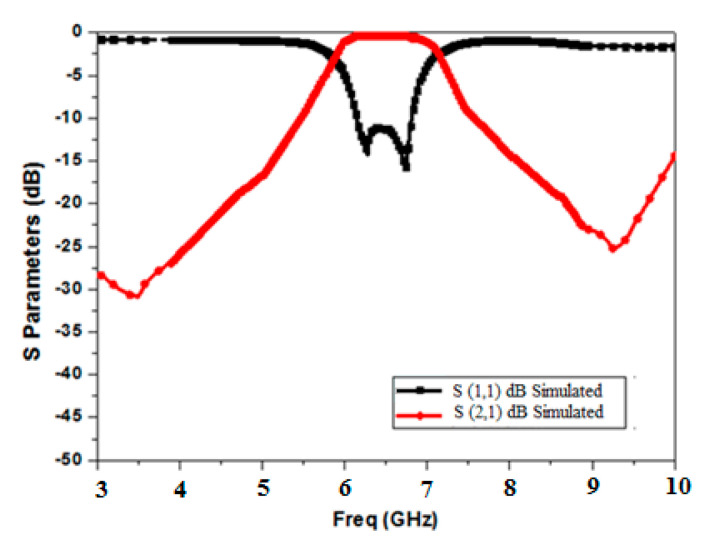
BPF frequency response of the comb-slotted HMSIW.

**Figure 6 micromachines-15-01440-f006:**
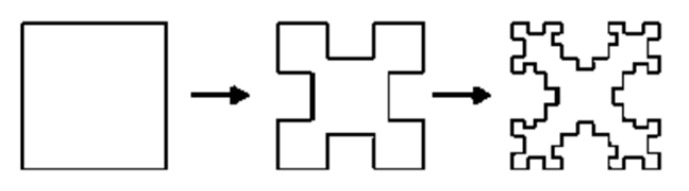
First three iterations of the Minkowski fractal curve [22].

**Figure 7 micromachines-15-01440-f007:**
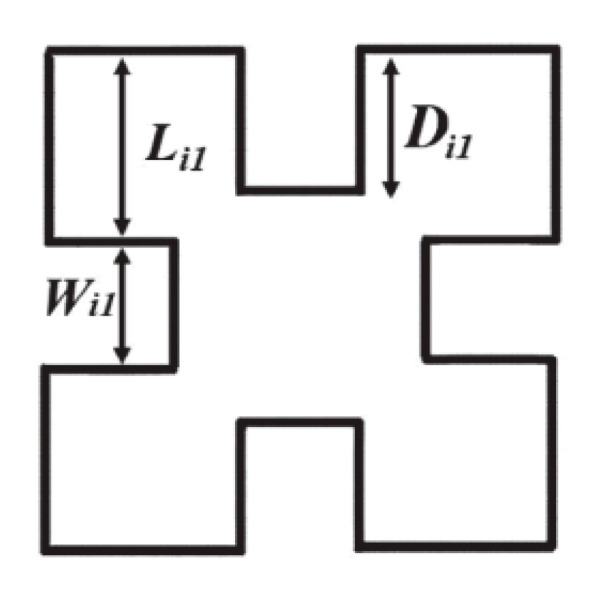
First iteration of the Minkowski curve (Li1 = Wi1 = 1.45 mm, Di1 = 1.25 mm).

**Figure 8 micromachines-15-01440-f008:**
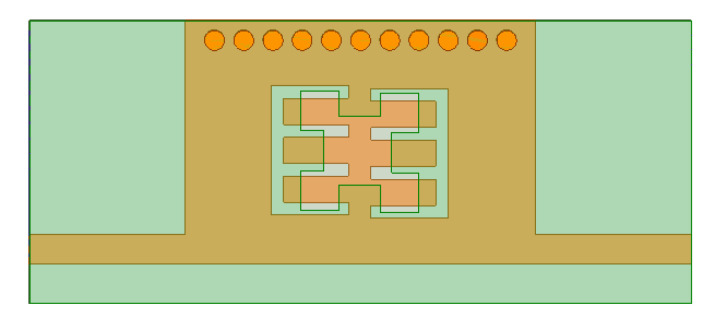
BPF using the Minkowski curve’s first iteration level as DGS.

**Figure 9 micromachines-15-01440-f009:**
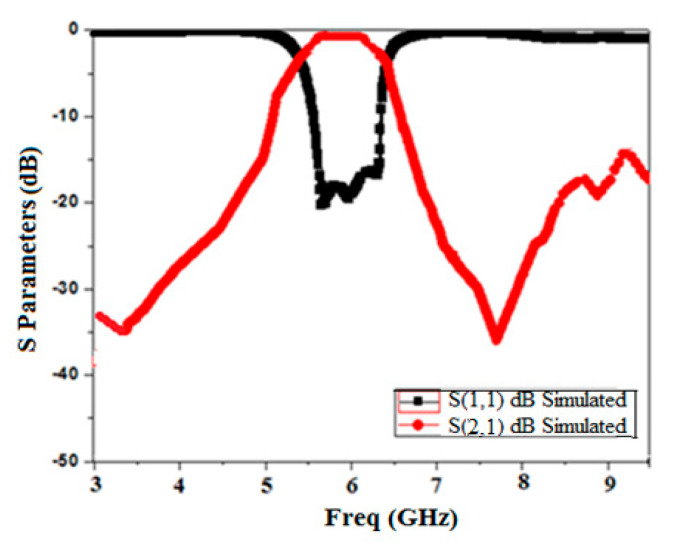
BPF frequency response based on the Minkowski curve’s first iteration level as the DGS.

**Figure 10 micromachines-15-01440-f010:**
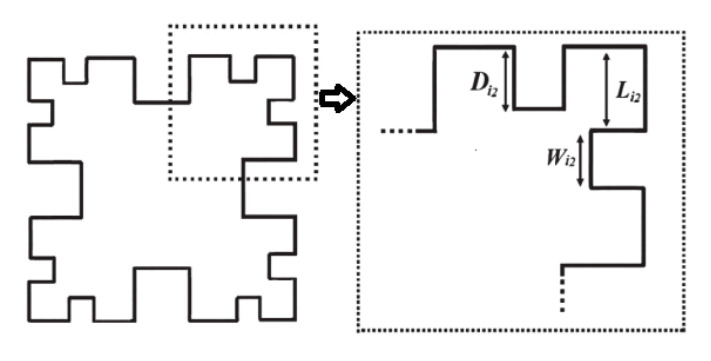
Second iteration of the Minkowski curve (Li2 = Wi2 = 0.45 mm, Di2 = 0.37 mm).

**Figure 11 micromachines-15-01440-f011:**
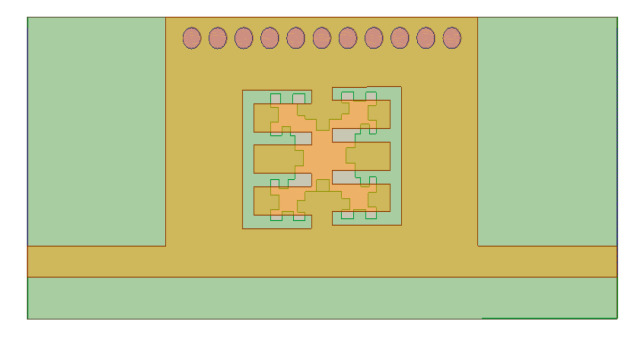
BPF based on the second iteration level of the Minkowski curve.

**Figure 12 micromachines-15-01440-f012:**
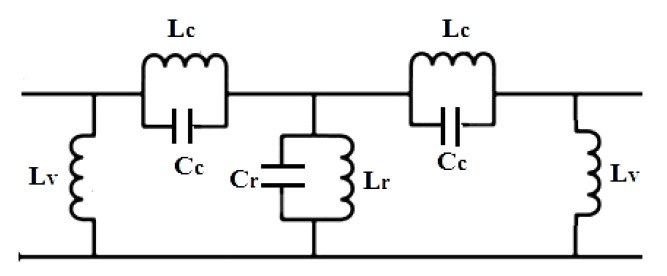
Equivalent circuit of the proposed filter. The element values are Lc = 0.12 nH, Cc = 4.5 pF, Lv = 1.88 nH, Lr = 0.18 nH, and Cr = 7.22 pF.

**Figure 13 micromachines-15-01440-f013:**
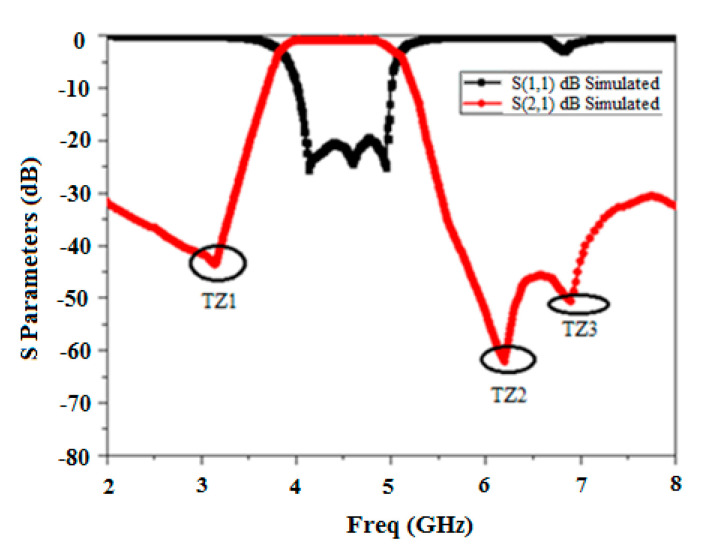
BPF frequency response based on the Minkowski curve’s second iteration level as the DGS.

**Figure 14 micromachines-15-01440-f014:**
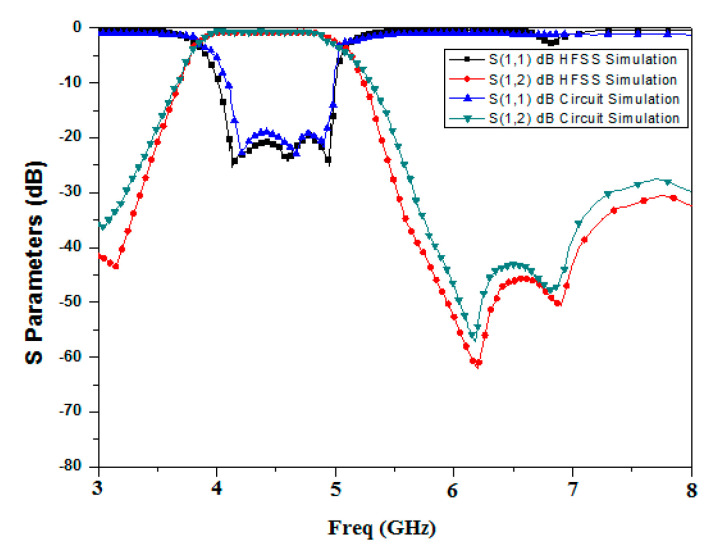
S parameters of EM simulation and circuit simulation.

**Figure 15 micromachines-15-01440-f015:**
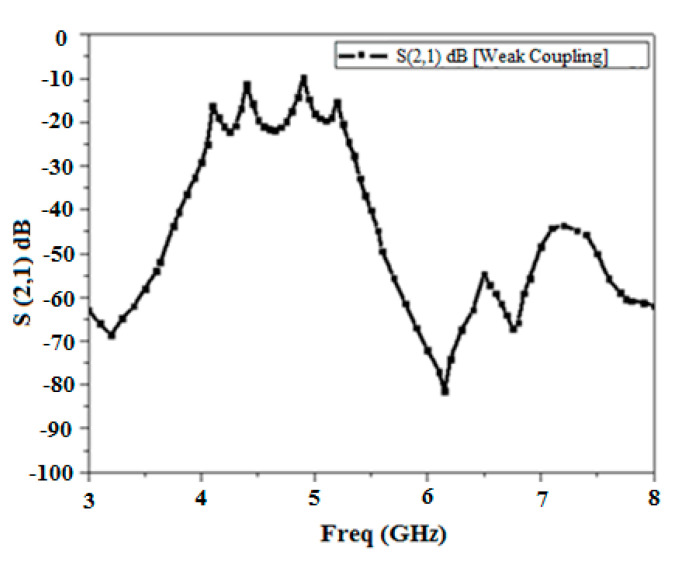
Simulated S21 response of a comb-slotted MMR filter with a DGS under weak coupling.

**Figure 16 micromachines-15-01440-f016:**
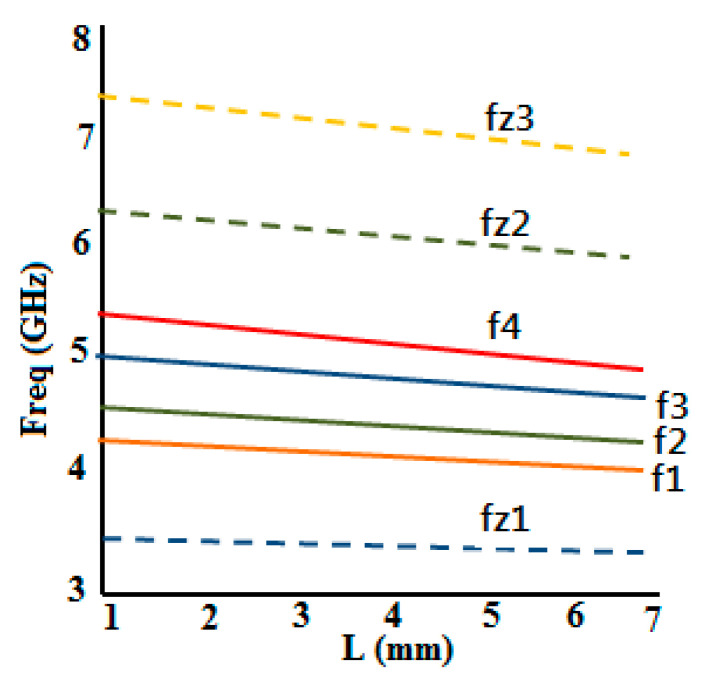
Mode chart of a comb-slotted MMR filter with a DGS for various values of L.

**Figure 17 micromachines-15-01440-f017:**
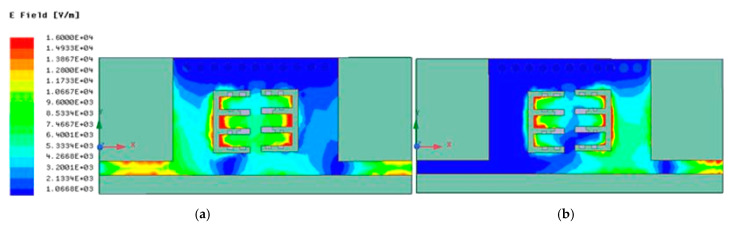
Current distribution in the (**a**) passband and (**b**) stopband.

**Figure 18 micromachines-15-01440-f018:**
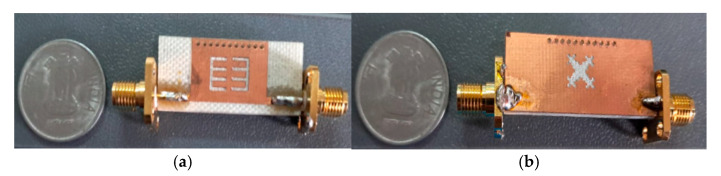
(**a**) Upper and (**b**) bottom views of the fabricated bandpass filter.

**Figure 19 micromachines-15-01440-f019:**
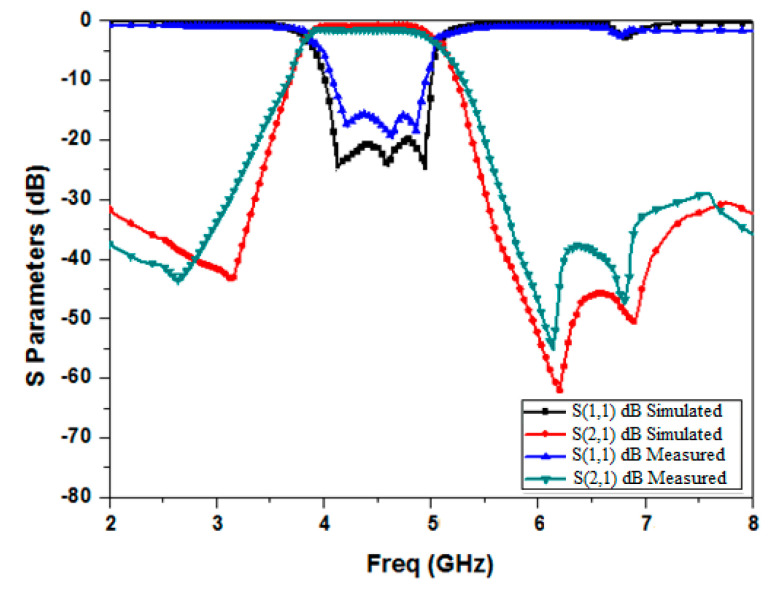
Comparison of the simulated and measured S parameter results.

**Figure 20 micromachines-15-01440-f020:**
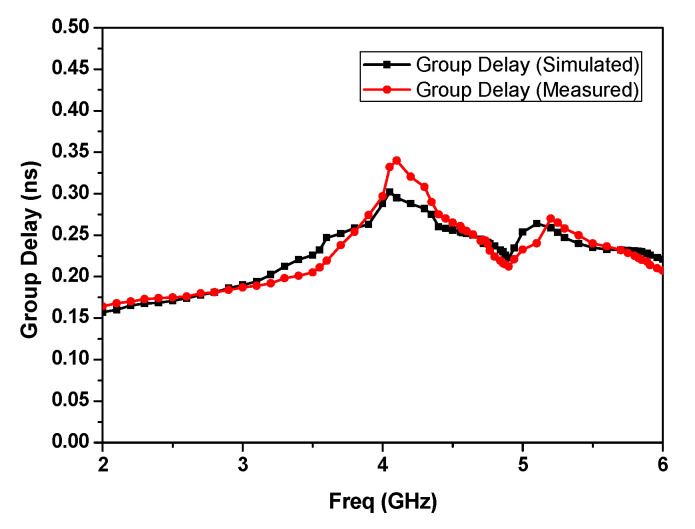
Comparison of the simulated and measured group delays.

**Table 1 micromachines-15-01440-t001:** Performance evaluation with contemporarily reported SIW BPFs working in C-band applications.

Ref. No.	Center Freq. f_o_ (GHz)	IL (dB)/RL (dB)	3 db-FBW (%)	Avg. Roll Off Rate (dB/GHz)	Out-of-Band Rejection (>−20 dB)	Q Factor	Area Size in mm^2^ (in λ_0_^2^)
[4]	6.11	1.1/20.5	8.9	45	3f_0_	58.5	25.5 × 9.6 (0.086)
[5]	7.15	2/25	1.55	28.4	Not Given	45.4	13.5 × 40 (0.32)
[6]	5.1	4/19	5.8	68.5	1.5f_o_	14.5	20 × 13 (0.075)
[7]	6.5	2.6/34	24.2	24.5	1.3f_o_	10.8	38.1 × 7.5 (0.21)
[8]	5.57	2/18	7.44	45.5	2f_o_	26.2	22.6 × 22.6 (0.16)
**[This work]**	**4.7**	**1.05/18.5**	**21.4**	**66.5**	**>2f_o_**	**11.4**	**19.5 mm × 6.5 mm** **(0.038)**

## Data Availability

The original contributions presented in the study are included in the article, further inquiries can be directed to the corresponding author.

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
