# Peer review of "Design of a Half-Mode Substrate-Integrated Waveguide (HMSIW) Multimode Resonator Bandpass Filter Using the Minkowski Fractal for C-Band Applications"

_micromachines, 2024, doi:10.3390/mi15121440_

Round 1
Reviewer 1 Report
Comments and Suggestions for Authors
The article describes the design of a compact and selective bandpass filter for C-band satellite communication applications. The filter improves performance by combining Minkowski fractal geometry as a defective ground structure (DGS) with a comb-shaped slotted Half-Mode Substrate Integrated Waveguide (HMSIW). The proposed design is simulated, constructed, and measured, revealing a small size, a wide stopband, and good selectivity.
1. Abstract
To support the study’s aims, the abstract focuses on designing a new bandpass filter based on HMSIW and Minkowski fractal DGS for C-band operation. The results are described comprehensively and accurately, and the stop-band performance and description of the resonance frequency are precise. However, the abstract could have mixed specifics about the achieved bandwidth and insertion loss.
Suggestions:
· Bandwidth should also be defined with specific numerical values to clarify the given issue and the results obtained by algorithms or analysis. Stop-band attenuation should also have specified numerical values.
· Further, mention the importance of using the Minkowski fractal to enhance performance.
· Use unique C-band application situations to provide context.
· Highlight what differentiates this approach from existing options.
2. Introduction
The paper's background introduces the topic by explaining the role of bandpass filters in satellite communication and the main benefits of applying SIW technology. While the literature review is strong at identifying prior work, there is limited comparison and contrast of the proposed approach to these methods. A precise presentation of the paper's contribution and novelty is also missing.
Suggestions:
- Unlike cited works, limitations are listed, and how the current work overcomes these limitations is described.
- State why the design that has been presented is, in fact, new (e.g., CMSIW, multimode resonator, and Minkowski fractal DGS are unique in this combination).
- It is advisable to provide a table showing a brief description and comparison in terms of size, performance, etc., in a case study of prior art.
3. Methodology
Design of Comb-Shaped Half Mode SIW (HMSIW):
This section introduces the design details of the basic HMSIW structure and comb-shaped slots. However, the dimensions are chosen to require deeper justification. While parametric analysis (Figure 3) is functional, the section could be improved by discussing the tradeoffs in selecting the optimal spacing (S).
Suggestions:
- Describe the HMSIW parameters in more detail, with a more detailed mention of the design equations used to determine these parameters.
- Explain why the parameters selected were chosen (why did they select the cut-off frequency of 8 GHz?).
- Discuss the advantages and disadvantages of the chosen design method and potential improvements.
Band Pass Filter Based on the First and Second Iterations of the Minkowski Fractal Curve:
The Minkowski fractal can be integrated as a DGS, as shown here. The fractal generation process is explained well but is a little too complex for a mainstream audience. The way the slow-wave effect is discussed and the bandwidth affected by parasitic capacitance are interesting. Still, a more rigorous analysis or simulation would have been an excellent addition. The equivalent circuit model (Figure 13) is helpful, but the component values from the derivation and validation have yet to be explained.
Suggestions:
- The Iteration Function System (IFS) and Minkowski fractal generation are explained simplistically.
- Analyses of the slow wave effect and its miniaturization impact.
- In comparison with simulation or measurement results, validate the equivalent circuit model.
4. Results and Findings
The fabrication process is described in this section, and a comparison between simulated and measured results is presented. For the most part, agreement between simulation and measurement is good, but there are variations, and these should be discussed along with possible causes (fab tolerances, measurement uncertainty). Although useful, this comparison with other state-of-the-art filters (Table 1) could be further expanded with more appropriate metrics and a more complete constellation of advantages and disadvantages for the proposed design.
Suggestions:
· To determine the differences between simulation and measurement, size the error sources, and conduct a more in-depth error analysis.
- Describe the advantages and disadvantages of using FR4 substrate and what other substrate material can benefit from using this substrate.
- Include additional performance metrics in expanded Table 1, such as Q factor and temperature stability.
5. Conclusion
Finally, conclusions rank the main findings and the merit of the proposed filter. However, the limitations of the work could be explicitly stated, and directions for future research could also be suggested.
Conclusion Suggestions:
- Limitations of the proposed design (e.g., performance at higher temperatures or different environmental conditions) are clearly stated.
- Identify specific areas for future directions (e.g., additional fractal geometries and the materials that support them or design improvement for specific uses).
- Compare the overall gained improvement with state of the art.
Technical and Language Corrections:
- Where technical terms are first used, ensure they are defined. For instance, definitions of Defected Ground Structure (DGS) and Half Mode Substrate Integrated Waveguide (HMSIW) bear clarification.
- Make sure that the words apply similarly across the paper. As an example, never use one type of which without explaining why, e.g., always use "Bandpass Filter" and not 'BPF' and 'bandpass filter' alternating.
· Make Ensure every reference is formatted correctly, following the journal's requirements.
Figures and Tables Suggestions:
- All figures must be of higher resolution with clear labels. Some additional annotations in Figures 1, 2, and 6 would be beneficial to clarify critical features further.
- To briefly explain each reference in Table 1 so that readers can understand how each reference compares to this work.
- A schematic figure of the filter design process or simulation setup is needed.
- Visual results can be compared more easily with bar charts or plots of insertion loss and return loss across different designs.
Additional Suggestions:
· Expand the literature review to include studies published more recently, detailing a more comprehensive set of recent studies that will be helpful to complement or compare with.
· The measurement setup and conditions under which the filter was tested are provided in more detail. Such a replication of the study would be helpful.
· In the discussion section, compare the proposed design and the current filters using performance metrics greater or equal to insertion loss and return loss.
· Finally, the findings suggest possible future research directions, such as other possible fractal geometries or materials for further improvement in filter performance.
· Provide a summary of the main conclusions and their practical implications for SATCOM or other domains to support the findings.
Comments on the Quality of English LanguageTechnical and Language Corrections:
- Where technical terms are first used, ensure they are defined. For instance, definitions of Defected Ground Structure (DGS) and Half Mode Substrate Integrated Waveguide (HMSIW) bear clarification.
- Make sure that the words apply similarly across the paper. As an example, never use one type of which without explaining why, e.g., always use "Bandpass Filter" and not 'BPF' and 'bandpass filter' alternating.
· Make Ensure every reference is formatted correctly, following the journal's requirements.
Figures and Tables Suggestions:
- All figures must be of higher resolution with clear labels. Some additional annotations in Figures 1, 2, and 6 would be beneficial to clarify critical features further.
- To briefly explain each reference in Table 1 so that readers can understand how each reference compares to this work.
- A schematic figure of the filter design process or simulation setup is needed.
- Visual results can be compared more easily with bar charts or plots of insertion loss and return loss across different designs.
Author Response
Please see the attachment for point wise response to reviewer 30 comments. Thanking you

Reviewer 2 Report
Comments and Suggestions for Authors
Dear authors,
After reading your manuscript, I have found it to be a paper presenting a well-conducted and valuable study that advances HMSIW filter designs with the novel application of Minkowski fractal DGS for C-band with demonstrations of a thoughtful balance of design innovation and experimental validation. The abstract is well written and covers most of the content of your manuscript. Here are some of my comments:
1) For the abstract, you should include a little bit more of your results. You only said the resonant frequency of BPF, and I think you can add the bandwidth and insertion loss results as well.
2) Line 32: don’t start with just a reference number. It’s awkward for readers and the sentence is not completed. Please follow line 34 for Ref 5 and change everyone that’s cited in this paragraph.
3) Line 33: define CSRR and later define BC-CSRR.
4) Line 45: repeated wording, delete the first “application”.
5) For section 2, when introducing the parameters, you used too many commas. For example, “HMSIW Length, LHM = 12 mm”, should just be “HMSIW Length LHM = 12 mm”. Please correct them.
6) Fig. 1 and Fig. 2 can be merged to just one figure with two panels.
7) For Fig. 3:
a. Fix the y-axis. It should be S(1,2) dB, not dB(S(1,2)). This also applies to other figures as well.
b. Why showing S12 not S21? I don’t understand and it’s not explained in the main text.
c. What are the lines connecting the data points? If they are just straight lines between two data points, then it’s not necessary to have them. You can keep them, but you should explain that these are not data or simulation results.
8) When specifying the bandwidth in Fig. 6, how did you determine it? 1 dB bandwidth or some other numbers?
9) Fig. 17 needs a scale bar for both panels.
10) Add size information about the coin you used in Fig. 18 so it can be used as a scale bar.
Thanks!
Author Response
Dear Esteemed Reviewer,
pls "Please see the attachment for pointwise reply to each comment. Hope it will satisfy you. Thanking you very much!! Best Regards

Round 2
Reviewer 1 Report
Comments and Suggestions for Authors
no more comment